# Catheter-Associated Urinary Infections and Consequences of Using Coated versus Non-Coated Urethral Catheters—Outcomes of a Systematic Review and Meta-Analysis of Randomized Trials

**DOI:** 10.3390/jcm11154463

**Published:** 2022-07-30

**Authors:** Vineet Gauhar, Daniele Castellani, Jeremy Yuen-Chun Teoh, Carlotta Nedbal, Giuseppe Chiacchio, Andrew T. Gabrielson, Flavio Lobo Heldwein, Marcelo Langer Wroclawski, Jean de la Rosette, Rodrigo Donalisio da Silva, Andrea Benedetto Galosi, Bhaskar Kumar Somani

**Affiliations:** 1Department of Urology, Ng Teng Fong General Hospital (NUHS), Singapore 609606, Singapore; vineetgaauhaar@gmail.com; 2Urology Unit, Azienda Ospedaliero-Universitaria Ospedali Riuniti di Ancona, Università Politecnica delle Marche, 60126 Ancona, Italy; carlottanedbal@gmail.com (C.N.); gipeppo1@gmail.com (G.C.); andreabenedetto.galosi@ospedaliriuniti.marche.it (A.B.G.); 3S.H.Ho Urology Centre, Department of Surgery, The Chinese University of Hong Kong, Hong Kong, China; jeremyteoh@surgery.cuhk.edu.hk; 4Brady Urological Institute, Johns Hopkins Medical Institutions, Baltimore, MD 21287, USA; andrewtgabe@gmail.com; 5Department of Urology, Universidade Federal de Santa Catarina, Florianópolis 88040-900, Brazil; flavio.lobo@gmail.com; 6Hospital Israelita Albert Einstein, São Paulo 05652-900, Brazil; urologia.marcelo@gmail.com; 7Beneficência Portuguesa de São Paulo (BP), São Paulo 01323-001, Brazil; 8Department of Urology, Medipol Mega University Hospital, Istanbul Medipol University, 34214 Istanbul, Turkey; j.j.delarosette@gmail.com; 9Division of Urology, Denver Health Medical Center, University of Colorado, Denver, CO 80204, USA; rodrigo.donalisiodasilva@dhha.org; 10Department of Urology, University Hospitals Southampton, NHS Trust, Southampton SO16 6YD, UK; bhaskarsomani@yahoo.com

**Keywords:** urinary catheters, catheters, indwelling, catheter-related infections

## Abstract

Coated urethral catheters were introduced in clinical practice to reduce the risk of catheter-acquired urinary tract infection (CAUTI). We aimed to systematically review the incidence of CAUTI and adverse effects in randomized clinical trials of patients requiring indwelling bladder catheterization by comparing coated vs. non-coated catheters. This review was performed according to the 2020 PRISMA framework. The incidence of CAUTI and catheter-related adverse events was evaluated using the Cochran–Mantel–Haenszel method with a random-effects model and reported as the risk ratio (RR), 95% CI, and *p*-values. Significance was set at *p* < 0.05 and a 95% CI. Twelve studies including 36,783 patients were included for meta-analysis. There was no significant difference in the CAUTI rate between coated and non-coated catheters (RR 0.87 95% CI 0.75–1.00, *p* = 0.06). Subgroup analysis demonstrated that the risk of CAUTI was significantly lower in the coated group compared with the non-coated group among patients requiring long-term catheterization (>14 days) (RR 0.82 95% CI 0.68–0.99, *p* = 0.04). There was no difference between the two groups in the incidence of the need for catheter exchange or the incidence of lower urinary tract symptoms after catheter removal. The benefit of coated catheters in reducing CAUTI risk among patients requiring long-term catheterization should be balanced against the increased direct costs to health care systems when compared to non-coated catheters.

## 1. Introduction

The word catheter is derived from the ancient Greek *kathiénai*, literally meaning “to thrust into” or “to send down” [1]. In use for more than 3500 years, urethral catheters are a bane and boon for patients and urologists alike as they may pose a risk to patients requiring long-term catheterization. The most common problems include hematuria, catheter encrustation requiring frequent catheter exchange, and catheter-acquired urinary tract infection (CAUTI).

With technical advancements in bioengineering and materials science, several types of indwelling catheters were developed to prevent CAUTI and improve patient tolerance. Coating agents were added to catheter surfaces to improve antimicrobial proprieties and to provide robust antibiofilm/antimicrobial activity, without causing an increase in patient discomfort [2,3]. Coated catheters can be divided into two types: those coated in antifouling materials, and those impregnated with bactericidal molecules.

Antifouling substances do not kill the bacteria but rather perturb their ability to colonize surfaces, preventing the formation of biofilms in the bladder or on the catheter surface. The most common antifouling materials are hydrogel and polytetrafluoroethylene (PTFE). Hydrogel catheters may reduce encrustation via forming hydration layers on the catheter surface; however, studies have demonstrated a similar incidence of nosocomial CAUTI and a higher rate of blockage when compared to standard silicone catheters [4]. PTFE-coated catheters seem to be more suitable candidates to inhibit biofilm formation because of their low coefficient of friction. Unfortunately, studies have demonstrated that PTFE-coated catheters are not superior to hydrogel or standard silicone catheters in preventing CAUTI [2].

Catheters can also be coated with antimicrobial agents such as metal ions (i.e., silver, gold, and/or palladium), antibiotics, and nitrofurazone. Among bactericidal-coated catheters, silver-coated catheters are the most popular and widely tested catheters. The release of silver ions into the bladder induces oxidative stress and disrupts bacteria membrane and proteins, but antimicrobial efficacy may vary with the silver-coated substance used. Although in vitro and in vivo studies have shown great efficacy in preventing infections [5], these have not necessarily translated to clear benefits in clinical trials [6].

Antibiotic-coated catheters are less frequently used, especially with the increased frequency of having multi-drug-resistant bacteria [2]. Nitrofurazone was a promising coating agent in in vivo and in vitro studies, but it was not efficient in preventing infections in clinical studies and caused patient discomfort [7].

This study aimed to systematically review the incidence of CAUTI and its adverse effects in randomized clinical trials of patients requiring indwelling bladder catheterization (transurethral or suprapubic) by comparing coated vs. non-coated catheters.

## 2. Materials and Methods

### 2.1. Aim of This Review

The present study aims to systematically review the incidence of CAUTI in patients requiring indwelling bladder catheterization by comparing coated vs. non-coated catheters. The primary outcome was the CAUTI rate between the two types of catheters. The secondary outcomes were the CAUTI rate according to catheterization time (cut-off: 14 days) and the rate of catheter-related adverse events (i.e., hematuria, need for catheter exchange or catheter removal, urinary symptoms after catheter removal). Additionally a cost-effectiveness analysis was performed.

### 2.2. Literature Search

This review was performed according to the 2020 Preferred Reporting Items for Systematic Reviews and Meta-Analyses (PRISMA) framework. A broad literature search was performed on 1 May 2022, using MEDLINE, EMBASE, and Cochrane Central Register of Controlled Trials. Medical Subject Heading (MeSH) terms and keywords such as (urinary tract infection OR infections OR sepsis) AND (short term OR long OR indwelling) AND (standard urethral catheter OR impregnated urethral catheter OR silicone OR hydrogel OR antibiotic coated OR silver-impregnated) were used. The search was restricted to English papers only. No date limits were imposed. Pediatric and animal studies were excluded. The review protocol was submitted for registration in PROSPERO (receipt #332889).

### 2.3. Selection Criteria

The Patient Intervention Comparison Outcome Study (PICOS) model was used to frame and answer the clinical question. P: adults requiring bladder catheterization; Intervention: coated catheters; Comparison: non-coated catheters; Outcome: CAUTI and catheter-related adverse effects; Study type: prospective and randomized studies. Patients were assigned to two groups according to the type of catheter (coated vs. non-coated catheters).

### 2.4. Study Screening and Selection

Two independent authors screened all retrieved records through Covidence Systematic Review Management^®^ (Veritas Health Innovation, Melbourne, Australia). Discrepancies were solved by a third author. Studies were included based on PICOS eligibility criteria. Only prospective and randomized studies were accepted. Meeting abstracts, retrospective, and prospective nonrandomized studies were excluded. Case reports, reviews, letters to the editor, and editorials were excluded. The full text of the screened papers was selected if found relevant to the purpose of this study.

### 2.5. Statistical Analysis

The incidence of CAUTI and catheter-related adverse effects was evaluated using the Cochran–Mantel–Haenszel method with a random-effects model and reported as the risk ratio (RR), 95% CI, and *p*-values. For studies with 3 groups of patients, intervention groups were combined to create a single pair-wise comparison [8]. Analyses were two tailed and significance was set at *p* < 0.05 and a 95% CI. Study heterogeneity was assessed utilizing the I^2^ value. Substantial heterogeneity was defined as an I^2^ value > 50%. Meta-analysis was performed using Review Manager (RevMan) 5.4 software by Cochrane Collaboration. The quality assessment of the included studies was performed using RoB 2 [9].

## 3. Results

The literature search retrieved 2689 studies. After eliminating 297 duplicates, 2392 studies were left for screening. Another 2326 papers were further excluded against the title and abstract screening because they were unrelated to the purpose of this review. The full texts of the remaining 66 studies were screened and 54 papers were further excluded. Finally, 12 studies were accepted and included for meta-analysis. Figure 1 shows the PRISMA flow diagram.

### 3.1. Study Characteristics and Quality Assessment

Twelve prospective, randomized studies compared coated vs. non-coated catheters in patients requiring an indwelling catheter [7,10,11,12,13,14,15,16,17,18,19,20]. No study with a suprapubic catheter was retrieved. Study characteristics are summarized in Table 1. Only one study had catheters with antibacterial/antifouling coating (i.e., hydrogel) [16] and the other 11 had catheters coated with bactericidal molecules, i.e., pure silver ions [7,10,12,13,18], noble ions (silver, gold, palladium) [14], or silver ions mixed with hydrogel [19], nitrofurazone [7,11,15,17], and a polymer of zinc oxide bonded carbon nanotube [20]. There were 36,783 patients included in 12 studies: 19,404 patients in the coated catheter group and 17,379 in the non-coated catheter group.

Table 2 shows data on pathogen species isolated in urine culture. The most common detected pathogens were *Escheria coli, Enterococcus*, *Pseudomonas* spp., *Klebsiella* spp., Gram-positive cocci, including *Staphylococcus aureus*, followed by *Candida* spp. and *Yeasts*. Polymicrobial infections were uncommon.

Figure 2 shows the details of quality assessment in the included studies. Six studies showed a low overall risk of bias and the remaining six demonstrated some concerns. The most common reason for bias arose from the randomization process, followed by bias due to missing outcome data.

### 3.2. Meta-Analysis of CAUTI

Meta-analysis from 12 studies (19,328 cases in the coated and 17,287 cases in the non-coated group) showed that the risk of CAUTI did not differ significantly between the groups (RR 0.87 95% CI 0.75–1.00, *p* = 0.06) (Figure 3). There was no significant heterogeneity among the studies (I^2^ = 22%). Subgroup analysis for catheter dwelling time demonstrated that the risk of CAUTI was significantly lower in the coated group compared with the non-coated group (RR 0.82 95% CI 0.68–0.99, *p* = 0.04). Only one study reported the rate of sepsis and another the rate of cystitis, making meta-analysis not feasible.

### 3.3. Meta-Analysis of Need for Catheter Removal or Catheter Exchange

Meta-analysis from three studies (499 cases in the coated and 502 cases in the non-coated group) showed no significant risk in the need for catheter removal or exchange (OR 0.93 95% CI 0.52–1.65, *p* = 0.80) (Figure 4). There was no significant heterogeneity among the studies (I^2^ = 0%).

### 3.4. Meta-Analysis of Lower Urinary Tract Symptoms at Follow-Up after Removal of Catheter

Meta-analysis from four studies (4245 cases in the coated and 2419 cases in the non-coated group) showed that the number of patients complaining of lower urinary tract symptoms after catheter removal did not differ between the groups (OR 1.05 95% CI 0.87–1.17, *p* = 0.58) (Figure 5). There was no significant heterogeneity among the studies (I^2^ = 14%).

### 3.5. Meta-Analysis of Hematuria Incidence

There was only one study reporting hematuria, making meta-analysis not feasible.

## 4. Discussion

In our meta-analysis, we found no difference in the incidence of CAUTI in patients with coated and non-coated catheters even though subgroup analysis regarding dwelling time (short- vs. long-term catheterization) showed a significantly lower risk for CAUTI in patients using coated catheters (*p* = 0.04). The interest in developing catheters that can decrease the risk of CAUTI started in 1979 with Akyama and Okamoto, who were the first to describe a decreased risk for bacteria associated with coated urinary catheters [21]. Other studies reported only a “protective effect” of coated urinary tract catheters but these trials were performed with a small number of patients [19,22,23]. Thibon et al. evaluated the effects of coated catheters with hydrogel and silver salts on the incidence of hospital-acquired urinary tract infection and showed no protective effect of coated catheters [19]. With regard to studies that reported a significant reduction in CAUTI in patients on silver-alloy catheters [12,22,23], some methodological critiques were made to these studies as they were performed by randomizing the hospital unit instead of the individual patients, which could lead to bias since hospital units can differ significantly in terms of catheter placement technique, indwelling time, and patient comorbidities.

Another confounding factor in considering indwelling catheters and CAUTI risk is the surgical procedure performed. Ideally, catheters should be removed at the earliest possible time. The misconception that the use of antibiotic- or silver-coated catheters has better outcomes in patients undergoing urological procedures needing a short duration of catheterization was refuted in a study by Pickard et al. [7]. Likewise, Erickson et al. compared silicone- and hydrogel-coated latex catheters in men needing short-term postoperative bladder drainage after urethral surgeries and showed no absolute advantage for either type [16]. Menzies et al. compared nitrofurazone-coated and non-coated urinary catheters in kidney transplant recipients and did not find any difference in the rate of urinary tract infection (8% and 6.8%, *p* = 0.99) among the two groups [11]. Instead, the incidence of adverse events was more frequent in the nitrofurazone-impregnated silicone urinary catheter group (46.6% and 26.1%, *p* = 0.007) [11]. Tae et al. studied the incidence of CAUTI in patients who underwent radical cystectomy with an orthotopic neobladder for bladder cancer and received either a coated or conventional non-coated catheter for 2 weeks [20]. The incidence of CAUTI 2 weeks after radical cystectomy and orthotopic neobladder was 21.95% (case) and 27.27% (control), with no significant difference between the two groups. However, asymptomatic bacteriuria was significantly lower in the antibiotic-coated catheter group [20]. The authors concluded that the prevention of biofilm formation on coated catheters has the potential to prevent CAUTI. One explanation for why the CAUTI rate was similar between the groups is that the duration of catheterization was short for this cohort (2 weeks); as we demonstrated in our meta-analysis, coated catheters may only be of benefit during longer catheterization durations. When taken together, the results of the present meta-analysis (Figure 3) support the safety of using non-coated catheters in patients undergoing surgical procedures in which catheter duration is expected to be less than 14 days. For patients requiring long-term catheters, the use of coated catheters may lower the risk of CAUTI together with routine catheter and/or drainage bag changes [24].

In a randomized trial of 17 patients, Priefer et al. observed that the practice of monthly catheter exchange resulted in fewer symptomatic urinary tract infections when compared to patients in whom catheters were exchanged at the time of either obstruction or infection [25]. In contrast, White et al. found that when patients were divided into short- versus long-term catheter exchange intervals, the incidence of infection was greater in those whose catheters were changed in 2 weeks or less [26]. Only 15.4% remained free of infection after one month in this group, whereas 80% of those whose catheters were changed between 4 and 6 weeks remained free of infection after 6 weeks. The number of exchange and the number of nurses who performed the catheter exchange might have influenced the CAUTI risk. Indeed, there is insufficient evidence to assess the value of different policies for replacing long-term urinary catheters on patient outcomes [24]. We found that the incidence of CAUTI was decreased when maintained well even for a long duration (RR 0.82 95% CI 0.68–0.99, *p* = 0.04). Thus, maybe the implementation of protocols using coated catheters could be of interest to prevent encrustation, obstruction, and infection, and increase the intervals between changes.

Adverse events related to catheter use, such as hematuria, irritative lower urinary tract symptoms, or the need for catheter exchange or removal, were investigated as secondary endpoints in our study. Only one article classified the infections by differentiating into cystitis or urinary sepsis, preventing our analysis from evaluating these secondary outcomes. Furthermore, no studies comparing coated versus non-coated catheters evaluated rates of pyelonephritis. There were insufficient data to determine the relative influence of coated urinary catheters on hematuria. Hematuria, which was reported in only a single study, occurred in 18/243 (7.4%) patients in the silver alloy-coated catheter group and 20/246 (8.1%) patients using conventional catheters and this was not significantly different between groups [18]. Three studies involving a total of 1001 patients reported on the need for catheter removal or exchange. Overall, the need for urinary catheter exchange or removal was similar between non-coated and coated catheters [14,18,20]. In our analysis, four studies, which included 6664 patients, provided information on lower urinary tract symptoms (LUTS) after catheter removal [7,14,17,18]. LUTS ranged from 1.2% to 22% in the coated group and from 0.4% to 22.6% in the control group. Compared to standard urinary catheters, we found that the use of coated catheters did not significantly increase the risk of LUTS.

Salient to the discussion of comparing antibiotic- or alloy-coated catheters to conventional silicone/latex catheters is cost-effectiveness. Overall, four studies incorporate cost-effectiveness analyses [12,27,28,29]. Cost analyses can be further stratified into comparisons of cost among different catheters and their associated components as well as analyses incorporating both catheter costs as well as the estimated cost of consequent nosocomial urinary tract infections. The latter cost assessment can be challenging to perform as it may be difficult to delineate how much a CAUTI contributes to the length of hospital stay or utilization of hospital resources. Nonetheless, several studies have provided estimates of these costs.

In a 12-month randomized crossover trial comparing CAUTI rates in patients with silver alloy-coated versus non-coated catheters, the use of silver alloy-coated catheters was associated with a 2.5-fold higher direct material cost when compared to non-coated catheters [12]. However, when taking into account the estimated costs associated with CAUTI and associated sequela (i.e., bloodstream infection, upper tract involvement, need for intensive care unit stay) within their study population, the use of silver alloy-coated catheters yielded significant aggregate savings due to a reduction in CAUTI rates. The lower and higher estimate of cost savings were USD 14,000 and 500,000, respectively [12]. This finding was similarly demonstrated by Bologna et al., where the use of silver alloy-coated catheters was predicted to lead to superior cost savings over standard latex catheters [27]. However, this cost analysis was limited to a single institution, whose differential CAUTI rate between silver alloy-coated and standard silicone catheters significantly differed from that of the other four institutions included in the analysis. The authors also relied on estimates of cost savings by attributing CAUTI as a major driver of hospital and intensive care unit length of stay [27]. Importantly, a recent prospective crossover study comparing silver alloy-coated to standard silicone catheters demonstrated a 12% risk reduction against CAUTI with the use of silver alloy-coated catheters. This is contrary to a prior study that assumed a 30–40% relative reduction in the CAUTI rate with the use of silver alloy-coated catheters in their cost-effectiveness analyses [29]. Therefore, if the difference in the CAUTI rate between catheter types is modest, the cost savings with the use of silver alloy-coated catheters may be negated and may not outweigh the increased direct costs associated with these catheters [29].

In another large study involving 7102 patients admitted to NHS England hospitals, cost-effectiveness analysis demonstrated that nitrofurazone-coated catheters were the least costly [30]. When compared to nitrofurazone-coated catheters, PTFE and silver alloy-coated catheters cost on average USD 11 and 19 more, respectively. Based on their analysis, nitrofurazone-coated catheters had an approximately 70% chance of being a cost-saving and had an 84% chance of having an incremental cost per quality-adjusted life year [incremental cost-effectiveness ratio of < GBP 300,000 (USD 47,500), the willingness-to-pay threshold suggested by the UK National Institute of Health and Clinical Excellence] [30]. Conversely, silver alloy-coated catheters had a 0% chance of being cost-effective at all threshold values between GBP 0 and 50,000. Nonetheless, nitrofurazone-coated catheters were associated with greater patient discomfort and the cost-saving estimates were based on assumptions of large attribution of CAUTI as the main driver of the length of hospital stay. These results, therefore, do not provide robust evidence of cost-effectiveness for one catheter over another within a universal health care system [30].

When taken together, the use of metal alloy-coated or antibiotic-coated catheters may increase direct costs to health care systems when compared to standard silicone or latex catheters; however, it is unclear whether the risk reduction in the CAUTI rate (and associated health care utilization) outweighs this cost.

Our study has some limitations. This study precludes us from making absolute deductions on which coated catheters are better for minimizing CAUTI, and better clinical trials should address this in the future. We could deduce that patients with long-term indwelling catheters could be the ideal candidates for coated catheters and it is necessary to provide proper training to patients and caregivers for catheter maintenance. This could help optimize the cost-effectiveness for the patients as, from our results, due to paucity of information and likely variability in health care systems, it was difficult to make concrete conclusions on cost-effectiveness. Finally, there was no randomized clinical trial comparing coated vs. non-coated suprapubic catheters, considering that UTI incidence is not significantly different between urethral and suprapubic catheters in spinal cord injury and neurogenic bladder [31].

## 5. Conclusions

In this systematic review of randomized trials, we found that the use of indwelling coated catheters was not associated with a lower incidence of CAUTI and the need for removal/change of catheter compared to non-coated catheters. In addition, we also found no difference in lower urinary tract symptoms after catheter removal. However, the incidence of CUATI was significantly lower using silver alloy-coated catheters in patients who require more than 14 days of dwelling time. The utility of coated catheters to reduce CAUTI risk versus standard catheters must be balanced with differences in direct costs to patients and health care systems.

## Figures and Tables

**Figure 1 jcm-11-04463-f001:**
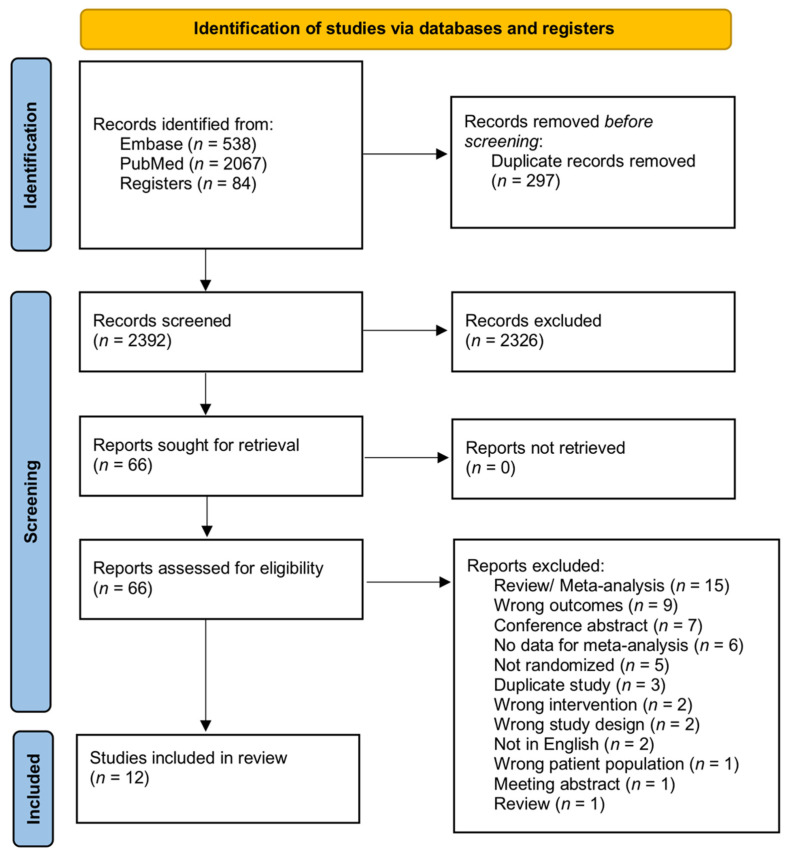
PRISMA diagram of this study.

**Figure 2 jcm-11-04463-f002:**
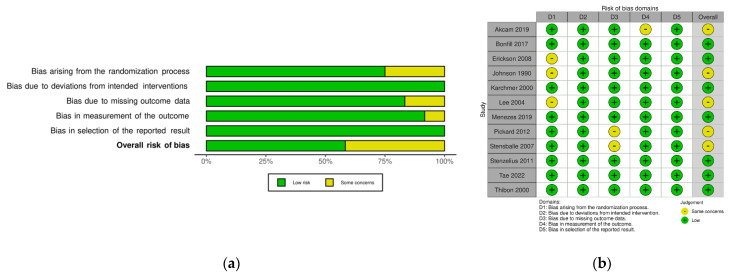
Risk of bias of the included study (Rob2): (**a**) review authors’ judgments about each risk of bias item presented as percentages across all included studies; (**b**) review authors’ judgments about each risk of bias item for each included study.

**Figure 3 jcm-11-04463-f003:**
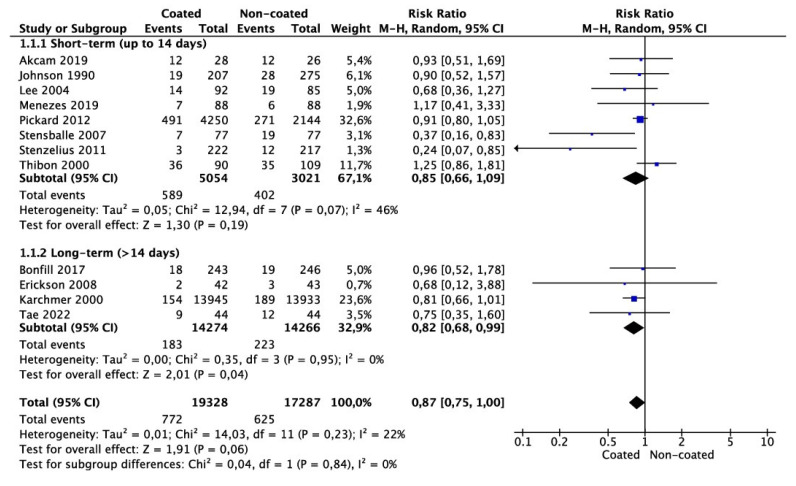
Meta-analysis of CAUTI incidence.

**Figure 4 jcm-11-04463-f004:**
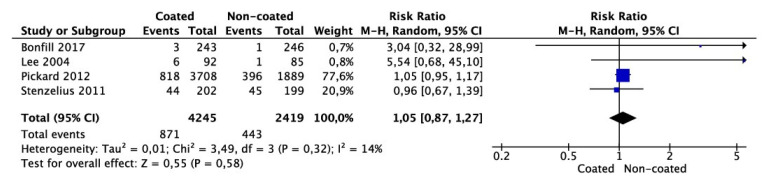
Meta-analysis of need for removal/change of catheter.

**Figure 5 jcm-11-04463-f005:**
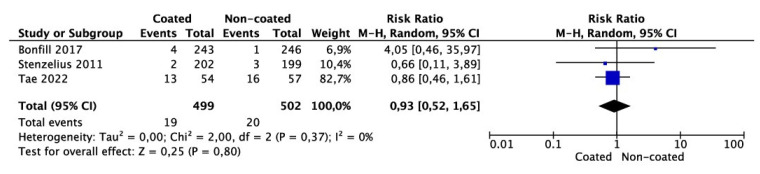
Meta-analysis of the number of patients reporting lower urinary tract symptoms at follow-up after removal of catheter.

**Table 1 jcm-11-04463-t001:** Characteristics of the included studies. **NA:** not available. **UTI:** urinary tract infections. **SCI**: spinal cord injury.

	Inclusion Criteria	Exclusion Criteria	Length of Follow Up	Type of Coated Catheter	Number of Patients Included in Coated Catheter	Mean Age (Standard Deviation) in Coated Group	Type of Non-Coated Catheter	Number of Patients Included in Non-Coated Catheter	Mean Age (Standard Deviation) in Non-Coated Group	Catheter Duration (Days)
Akcam 2019 [10]	Patients admitted to the intensive care unit and anticipated to require long-term urinary catheterization	Patients with any infectious disease on admission or with pyuria/bacteriuria in the first urine specimen collected following catheter placement	Until discharge of patients	Silver-coated silicone catheters	28	70.61 (NA)	silicone catheters	26	69.23 (NA)	NA
Bonfill 2017 [18]	Patients with traumatic or medical SCI requiring an indwelling urinary catheter for at least 7 days	Patients with demonstrable UTI at the time of inclusion; taking antibiotic treatment at the time of inclusion for any infectious condition or within 7 days before inclusion	12 months	Silver alloy catheters	243	55.30 (16.35)	silicone catheters	246	57.25 (16.32)	27 in coated 28 in non-coated
Erickson 2008 [16]	Men undergoing urethral reconstruction	None	20 months	Hydrogel-coated latex foley	42	40 (NA)	silicone catheters	43	43 (NA)	14–21
Johnson 1990 [13]	Patients with a steady catheter that was expected to remain indwelling for at least 24 h	UTI at enrollment	16 months	Silicone catheter coated with a layer of silicone elastomer containing micronized silver oxide	207	50 (NA)	silicone catheters	208	47 (NA)	3 in coated 4 in non-coated
Karchmer 2000 [12]	Hospitalized patients with vesical catheters	Pediatric, obstetrics, gynecology, and psychiatry wards excluded	12 months	Silver-alloy, hydrogel-coated latex catheters	13,945	NA	silicone catheters	13,933	NA	>7 days
Lee 2004 [17]	Patients who were catheterized for more than 24 h	conditions such as silicone sensitivity, nitrofurazone or nitrofurantoin sensitivity, pregnancy, lactating, hospitalization for more than 7 days, and having urinary diseases; positive urine culture result before catheter insertion or when the catheter was removed within 24 h of insertion	7 days	Release nitrofurazone foley catheter	92	55.3	silicone catheters	85	54.1	3.9–4.4
Menezes 2019 [11]	urethral catheterization for kidney transplantation with a living donor	asymptomatic bacteriuria or urinary tract infection at baseline, deceased kidney transplant donors, hypersensitivity to nitrofurantoin, pregnancy	22 months	Nitrofural-impregnated silicone catheter	88	38.4 (NA)	silicone catheters	88	39.6 (NA)	5.1
Pickard 2012 [7]	Adults undergoing urethral catheterization for an anticipated duration of up to 14 days (including people with diabetes and individuals treated with immunosuppressive drugs)	Symptomatic urinary tract infection at baseline, urological procedures in the previous 7 days, or allergies to catheter materials	39 months	(1) Silver alloy-coated latex catheter (2) Nitrofural-impregnated silicone catheter	(1) 2097 (2) 2153	(1) 59 (16) (2) 59 (16)	standard polytetrafluoroethylene (PTFE)-coated latex catheter	2144	59 (16)	2 (1–3)
Stensballe 2007 [15]	trauma patients who needed a urinary catheter and were admitted directly from the accident scene to the Trauma Center	HIV infection; preinjury treatment with corticosteroids; pregnancy; primary burn injury; and unattainable signed informed consent	24 months	Nitrofurazone-impregnated catheter	106	41 (NA)	silicone catheters	106	43 (NA)	2 (0–7)
Stenzelius 2011 [14]	patients undergoing elective orthopedic surgery	recent (within 3 weeks) use of a urinary catheter or a recent history of UTI, previous radiation therapy over the lower pelvis, latex allergy, cognitive impairment, or difficulties in understanding the Swedish language	2–7 days	Noble metal alloy-coated latex catheter	222	67.6 (12)	silicone catheters	217	66.7 (12.8)	2
Tae 2022 [20]	Patients underwent radical cystectomy with neobladder cause of invasive bladder cancer	Malnutrition, active infection, immunodeficiency, allergy to components	NA	Carbon nanotube and ZnO-bonded CNT	41	65.22 (10.25)	silicone catheters	44	65.36 (8.56)	14 + or − 1
Thibon 2000 [19]	Patients in neurosurgery ICU required catheter during admission for more than three days and had to stay in hospital for at least10 days after the insertion of a urinary catheter	urinary tract infection or inflammation of the perineum or penis before catheter insertion, allergy to hydrogel or silver salts, contraindications for catheterization, urinary tract catheter insertion during the 48 h before inclusion, antibiotic treatment for urinary tractinfection and other types of urinary tract intervention (prostate, bladder)	24 months	Hydrogel and silver salt-coated catheter	90	59.8 (17.1)	silicone catheters	109	60.5 (15.5)	10

**Table 2 jcm-11-04463-t002:** Pathogens isolated in urine cultures.

	Pathogen Species in Urine Culture	Difference in Urine Culture between Coated and Non-Coated Catheters
Akcam 2019 [10]	The most commonly detected agent, at 11/25 (44%), was Escherichia coli (44%), *Enterococcus* spp. (20%), Klebsiella pneumonia (8%), *Pseudomonas* spp. (8%), *Acinetobacter* spp. (8%), *Enterobacter cloacae* (4%), Proteus mirabilis (4%) and *Candida* spp. (4%). Second species were grown in four of the specimens: *Enterococcus* spp. was isolated in three specimens, and *E. cloacae* in one	*E. coli* grew in 26.9% and microorganisms other than *E. coli* in 19.3% of the subjects using normal catheters, while *E. coli* grew in 14.3% and other microorganisms in 32.1% of the patients using silver-coated catheters
Bonfill 2017 [18]	Not reported in full	One patient, pertaining to the group with SAC urinary catheter, developed a urinary septic shock caused by Proteus mirabilis. Another patient, pertaining to the group of standard urinary catheter, developed a urinary sepsis caused by *Escherichia coli* and *P. mirabilis*
Erickson 2008 [16]	Not reported	Not reported
Johnson 1990 [13]	*Coagulase-negative staphylococci*, *Enterococcus species*, *Escherichia coli*, *Proteus mirabilis*, *Pseudomonas* species, Yeast other	No difference
Karchmer 2000 [12]	*Escherichia coli* (18.4%), *Escherichia faecalis* (16.9%), *Candida albicans* (13.4%), and *Pseudomonas aeruginosa* (11.7%), *Yeast* (26.2%), *Gram-positive cocci*, including *Staphylococcus aureus*, *coagulase-negative staphylococci*, and *enterococci* (28%)	There were no statistically significant differences in the proportion of infections attributed to different organisms following use of silver-coated and uncoated catheters
Lee 2004 [17]	*Enterococcus species* (22.5%), *Staphylococcus* (15%), *Pseudomonas species* (30%), *Stenotrophomonas**Maltophilia* (10%), others (*Acinetobacter calcoaceticus–baumannii* complex, *A. lwoffi*, *Citrobacter freundii*, *Enterobacter cloacae*, *Nonfermenting Gram-negative Bacillus*, *Pasteurella multocida*, *Burkholderia cepacia*, *B. pseudomallei*, *Chryseobacterium meningosepticum* 15%). Mixed infection was observed in five patients	*Stenotrophomonas**Maltophilia* was not isolated in patients with non-coated catheters
Menezes 2019 [11]	*Gram-negative bacilli* (95.24%) and *Escherichia coli* was the most frequently isolated microorganism (47.62%). Among the isolates of *E coli* and *Klebsiella pneumoniae*, 25% had an extended spectrum beta-lactamase producing profile, and 12.5% of the *K pneumonie* strains were carbapenem resistant	No evidence of enhanced antimicrobial resistance with the employment of the Nitrofurazone-coated urinary catheter
Pickard 2012 [7]	Not reported	Not reported
Stensballe 2007 [15]	*Enterococcus species*, *Escherichia coli*, *Candida species*, *Coagulase-negative staphylococci*, *Corynebacterium species*, *Pseudomonas aeruginosa*, *polymicrobial*	Nitrofurantoin resistance was found in 3 isolates in the nitrofurazone group (1 with *Pseudomonas aeruginosa* and 2 with *Candida species*) compared with 7 in the silicone group (1 with *Enterobacter species*, 5 with *Candida species,* and 1 with *Enterobacter species* and *Candida species*)
Stenzelius 2011 [14]	Not reported	Not reported
Tae 2022 [20]	*Enterococcus faecalis*, *Pseudomonas aeruginosa*, *Yeast*, *Streptococcus species*, *Klebsiella pneumoniae*, *methicillin-resistant Staphylococcus aureus*	Coated: 19 positive cultures. Non-coated: 22 positive cultures *Enterococcus faecalis*: coated 8; non-coated 11 *Pseudomonas aeruginosa*: coated 4; non-coated 4 *Yeast:* coated 3; non-coated 2 *Streptococcus species*: coated 2; non-coated 4 *Klebsiella pneumoniae*: coated 1; non-coated 2 *methicillin-resistant Staphylococcus aureus*: coated 1; non-coated 1
Thibon 2000 [19]	*Escherichia coli*, *Proteus*, *Pseudomonas, Enterobacter cloacae*, *Yeasts*, *coagulase negative staphylococci*, *enterococci*, others	There was no significant difference between the types of organism identified with the two types of catheter

## Data Availability

Data will be provide by the corresponding author upon a reasonable request.

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
