# Peer review of "Catheter-Associated Urinary Infections and Consequences of Using Coated versus Non-Coated Urethral Catheters—Outcomes of a Systematic Review and Meta-Analysis of Randomized Trials"

_jcm, 2022, doi:10.3390/jcm11154463_

Round 1
Reviewer 1 Report
Dear Authors,
The subject of the reviewed manuscript is very important, mainly due to the clinical aspect. The introduction section presents background information, this part is concise and well‐written. The methods presented in the study have been well chosen and described. The presentation of the results is correct. In the discussion section, the authors confront their observations with the findings of other researchers. The literature is closely related to the topic of the article.
Minor corrections in the manuscript text must be performed to increase its quality:
line 46 is:... tollerance.., should be:.. tolerance..
line 50 is:.. those with coated.., should be:.. those coated..
line 73 is:.. and adverse.., should be:.. and its adverse..
line 152 is:.. characteristics if the included studies.., should be:. Characteristics of the included studies..
Author Response
Dear Authors,
The subject of the reviewed manuscript is very important, mainly due to the clinical aspect. The introduction section presents background information, this part is concise and well‐written. The methods presented in the study have been well chosen and described. The presentation of the results is correct. In the discussion section, the authors confront their observations with the findings of other researchers. The literature is closely related to the topic of the article.
REPLY. We would like to thank you for these nice words about our manuscript.
Minor corrections in the manuscript text must be performed to increase its quality:
line 46 is:... tollerance.., should be:.. tolerance..
REPLY. We would like to thank you for highlighting this mistake. We do apologize for this oversight that has been corrected
line 50 is:.. those with coated.., should be:.. those coated..
REPLY. We would like to thank you for highlighting this mistake. We do apologize for this oversight that has been corrected
line 73 is:.. and adverse.., should be:.. and its adverse..
REPLY. We would like to thank you for highlighting this mistake. We do apologize for this oversight that has been corrected
line 152 is:.. characteristics if the included studies.., should be:. Characteristics of the included studies..
REPLY. We would like to thank you for highlighting this mistake. We do apologize for this oversight that has been corrected

Reviewer 2 Report
This review was very informative in its comparative study of coated and uncoated bladder urethral catheters.
Please comment on the following.
Major comments
1) Regarding the subject study, is it necessary to separate analyses of postoperative catheterization from studies of catheterization in chronic disease? In those cases, I think the development of CAUTI is very different depending on the presence or absence of the underlying disease. So I do not believe discussing them at the same table is appropriate. Also, risk factors for UTI include the length of catheterization, women, diabetes, open-closed system, and suboptimal aseptic technique. Are these taken into account to compare studies?
2) As for your conclusion, isn't the main thing that there is no significant difference between coated and uncoated catheters, and is the following an exaggeration? I felt there was not enough discussion on this conclusion.
"silver alloy-coated urinary catheters are better suited in patients where the indwelling time is more than 14 days to minimize CAUTI."
Minor comments
1) How much is the general price difference between coated and uncoated catheters? It seems to me that most are becoming coated now.
2) Are catheter manufacturers possibly sponsoring past studies to promote sales in these studies?
3) What do you think is the clinical significance of asymptomatic pyuria? Pyuria is used to evaluate the efficacy of CIC, but can it be considered a risk for UTI in the case of indwelling catheters?
4) Do you have any data on bacterial species in culture? For example, is it mainly indigenous skin bacteria?
Author Response
Reviewer #2
This review was very informative in its comparative study of coated and uncoated bladder urethral catheters.
REPLY. We would like to thank you for these nice words about our manuscript
Please comment on the following.
Major comments
- Regarding the subject study, is it necessary to separate analyses of postoperative catheterization from studies of catheterization in chronic disease? In those cases, I think the development of CAUTI is very different depending on the presence or absence of the underlying disease. So I do not believe discussing them at the same table is appropriate. Also, risk factors for UTI include the length of catheterization, women, diabetes, open-closed system, and suboptimal aseptic technique. Are these taken into account to compare studies?
REPLY. We would like to thank you for this comment. We agree with you that the development of CAUTI also depends on comorbidity open-closed system and suboptimal aseptic technique. However, we were not able to perform different sub-analyses (e.g. according to gender and chronic disease) because data on CAUTI was reported altogether as the total number with no differentiation among gender or comorbidity. We decided to keep all studies in Table 1. The main goal of Table 1 is simply reporting inclusion and exclusion criteria, types of catheters used and the number of patients enrolled in each study. We think that this Table would give the journal readership an overview of included studies.
2) As for your conclusion, isn't the main thing that there is no significant difference between coated and uncoated catheters, and is the following an exaggeration? I felt there was not enough discussion on this conclusion.
"silver alloy-coated urinary catheters are better suited in patients where the indwelling time is more than 14 days to minimize CAUTI."
REPLY. We would like to thank you for this comment. The conclusion has been rephrased and implemented as follows “In this systematic review of randomized trials, we found that the use of indwelling coated catheters was not associated with a lower incidence of CAUTI and the need for removal/change of catheter compared to non-coated catheters. In addition, we also found no difference in lower urinary tract symptoms after catheter removal. However, the incidence of CUATI was significantly lower using silver alloy-coated in patients who require more than 14 days of dwelling time. The utility of the coated catheters to reduce CAUTI risk versus standard catheters must be balanced with differences in direct costs to patients and healthcare systems.”
Minor comments
1) How much is the general price difference between coated and uncoated catheters? It seems to me that most are becoming coated now.
REPLY. We would like to thank you for this important comment. There are no data on the general price difference between coated and uncoated catheters in the included studies. Prices may vary across Countries and change year by year. In addition, considering only the price of a single catheter in a patient does make no sense because the main goal is the cost-benefit analysis. Therefore, a cheaper catheter might be associated with a higher incidence of CAUTI, adverse events with a longer hospital stay, and unplanned changes in a patient making it finally “more expensive”. That is why the cost-benefit analysis in four studies in our manuscript did not show a net benefit of one type of catheter over the other one.
2) Are catheter manufacturers possibly sponsoring past studies to promote sales in these studies?
REPLY. We would like to thank you for this important comment. Among the studies included in our analysis, only 3 studies reported that their projects were supported by manufacturers which provided catheters (i.e. references #20, #14, and #17 in our study). However, the authors of these studies declared that manufacturers were not involved in their study because the design, data collection, and analysis have been done totally without any insight from manufacturers. Therefore, we believe that sponsoring companies did not influence the results of those 3 studies.
3) What do you think is the clinical significance of asymptomatic pyuria? Pyuria is used to evaluate the efficacy of CIC, but can it be considered a risk for UTI in the case of indwelling catheters?
REPLY. We would like to thank you for this important comment. Pyuria has been shown to have excellent predictive value for identifying urinary tract infections in non-catheterized patients (Mabeck CE. Studies in urinary tract infections, IV: urinary leukocyte excretion in bacteriuria. Acta Med Scand. 1969;186:193-198. 39. Stamm WE, et al. Treatment of the acute urethral syndrome. N Engl J Med. 1981;304:956-958.) but pyuria is less strongly correlated with CAUTI in patients with short-term indwelling urinary catheters (Tambyah PA, Maki DG. The relationship between pyuria and infection in patients with indwelling urinary catheters: a prospective study of 761 patients. Arch Intern Med. 2000 Mar 13;160(5):673-7.). In patients with long-term indwelling urinary catheters, pyuria has even less important because the routine change of chronic catheters in asymptomatic individuals causes a significant rise in the urinary white blood cells, and pyuria and bacteriuria are prevalent among individuals with indwelling Foley catheters, including those who are asymptomatic (Ho CH, et al.. Effects of the routine change of chronic indwelling Foley catheters in persons with spinal cord injury. J Spinal Cord Med. 2001 Summer;24(2):101-4.; Tambyah PA, Maki DG. Catheter-associated urinary tract infection is rarely symptomatic: a prospective study of 1,497 catheterized patients. Arch Intern Med. 2000 Mar 13;160(5):678-82)
4) Do you have any data on bacterial species in culture? For example, is it mainly indigenous skin bacteria?
REPLY. We would like to thank you for this important comment. We looked at pathogens in each study and built the following Table that includes all species isolated. We added this table in the results section.
|
|
Pathogen species in urine culture |
Difference in urine culture between coated and non-coated catheters |
|
Akcam 2019
|
The most commonly detected agent, at 11/25 (44%), was Escherichia coli (44%), Enterococcus spp. (20%), Klebsiella pneumonia (8%), Pseudomonas spp. (8%), Acinetobacter spp. (%8), Enterobacter cloacae (4%), Proteus mirabilis (4%) and Candida spp. (4%). Second species were grown in four of the specimens: Enterococcus spp. was isolated in three specimens, and E. cloacae in one |
E. coli grew in 26.9% and microorganisms other than E. coli in 19.3% of the subjects using normal catheters, while E. coli grew in 14.3% and other microorganisms in 32.1% of the patients using silver-coated catheters. |
|
Bonfill 2017
|
Not reported in full |
One patient, pertaining to the group with SAC urinary catheter, developed a urinary septic shock caused by Proteus mirabilis. Another patient, pertaining to the group of standard urinary catheter, developed urinary sepsis caused by Escherichia coli and P. mirabilis |
|
Erickson 2008
|
Not reported |
Not reported |
|
Johnson 1990
|
Coagulase-negative staphylococci, Enterococcus species, Escherichia coli, Proteus mirabilis, Pseudomonas species, Yeast other |
No difference |
|
Karchmer 2000
|
Escherichia coli (18.4%), Escherichia faecalis (16.9%), Candida albicans (13.4%), and Pseudomonas aeruginosa (11.7%), Yeast (26.2%), Gram-positive cocci, including Staphylococcus aureus, coagulase-negative staphylococci, and enterococci (28%)
|
There were no statistically significant differences in the proportion of infections attributed to different organisms following use of silver-coated and uncoated catheters |
|
Lee 2004
|
Enterococcus species (22.5%), Staphylococcus (15%), Pseudomonas species (30%), Stenotrophomonas Maltophilia (10%), others (Acinetobacter calcoaceticus–baumannii complex, A. lwoffi, Citrobacter freundii, Enterobacter cloacae, Nonfermenting Gram-negative Bacillus, Pasteurella multocida, Burkholderia cepacia, B. pseudomallei, Chryseobacterium meningosepticum 15%). Mixed infection was observed in five patients |
Stenotrophomonas Maltophilia was not isolated in patients with non-coated catheters |
|
Menezes 2019
|
Gram‐negative bacilli (95.24%) and Escherichia coli was the most frequently isolated microorganism (47.62%). Among the isolates of E coli and Klebsiella pneumoniae, 25% had an extended spectrum beta‐lactamase producing profile, and 12.5% of the K pneumonie strains were carbapenem‐resistant |
No evidence of enhanced antimicrobial resistance with the employment of the Nitrofurazone‐coated urinary catheter |
|
Pickard 2012
|
Not reported |
Not reported |
|
Stensballe 2007
|
Enterococcus species, Escherichia coli, Candida species, Coagulase-negative staphylococci, Corynebacterium species, Pseudomonas aeruginosa, polymicrobial |
Nitrofurantoin resistance was found in 3 isolates in the nitrofurazone group (1 with Pseudomonas aeruginosa and 2 with Candida species) compared with 7 in the silicone group (1 with Enterobacter species, 5 with Candida species, and 1 with Enterobacter species and Candida species). |
|
Stenzelius 2011
|
Not reported |
Not reported |
|
Tae 2022
|
Enterococcus faecalis, Pseudomonas aeruginosa, Yeast, Streptococcus species, Klebsiella pneumoniae, methicillin-resistant Staphylococcus aureus. |
Coated: 19 positive cultures. Non-coated: 22 positive cultures Enterococcus faecalis: coated 8; non-coated 11 Pseudomonas aeruginosa: coated 4; non-coated 4 Yeast: coated 3; non-coated 2 Streptococcus species: coated 2; non-coated 4 Klebsiella pneumoniae: coated 1; non-coated 2 methicillin-resistant Staphylococcus aureus: coated 1; non-coated 1 |
|
Thibon 2000
|
Escherichia coli, Proteus, Pseudomonas, Enterobacter cloacae, Yeasts, coagulase negative staphylococci, enterococci, others |
There was no significant difference between the types of organism identified with the two types of catheters. |

Round 2
Reviewer 2 Report
The authors have adequately answered some of my questions and have made additions and changes to the paper. Therefore, I have no further particular comments.